# Remaining Useful Life Prediction Method for Bearings Based on LSTM with Uncertainty Quantification

**DOI:** 10.3390/s22124549

**Published:** 2022-06-16

**Authors:** Jinsong Yang, Yizhen Peng, Jingsong Xie, Pengxi Wang

**Affiliations:** 1School of Traffic and Transportation Engineering, Central South University, Changsha 410083, China; yangjs@csu.edu.cn (J.Y.); jingsongxie@csu.edu.cn (J.X.); 2College of Mechanical and Vehicle Engineering, Chongqing University, Chongqing 400030, China; 3School of Mechanical Engineering, Xi’an Jiaotong University, Xi’an 710049, China; wangpengxi@stu.xjtu.edu.cn

**Keywords:** remaining useful life (RUL), degradation feature screening, LSTM, uncertainty

## Abstract

To reduce the economic losses caused by bearing failures and prevent safety accidents, it is necessary to develop an effective method to predict the remaining useful life (RUL) of the rolling bearing. However, the degradation inside the bearing is difficult to monitor in real-time. Meanwhile, external uncertainties significantly impact bearing degradation. Therefore, this paper proposes a new bearing RUL prediction method based on long-short term memory (LSTM) with uncertainty quantification. First, a fusion metric related to runtime (or degradation) is proposed to reflect the latent degradation process. Then, an improved dropout method based on nonparametric kernel density is developed to improve estimation accuracy of RUL. The PHM2012 dataset is adopted to verify the proposed method, and comparison results illustrate that the proposed prediction model can accurately obtain the point estimation and probability distribution of the bearing RUL.

## 1. Introduction

The remaining useful life (RUL) prediction for rolling bearings is of great significant and practical value for the predictive maintenance of mechanical equipment and safe operation of industries [1,2]. In recent years, many scholars have studied and summarized RUL prediction methods [3,4,5]. With the gradual accumulation of industrial condition monitoring data and the continuous improvement of computer computing power, data-driven RUL prediction methods have become the focus of prognostic and health management (PHM) research. Whereas, the rich condition monitoring data led to formidable challenges to traditional artificial intelligence methods [6]. Deep learning methods have powerful information extraction capabilities and can quickly extract effective information from massive data. Therefore, RUL prediction methods based on deep learning are favored by an increasing number of scholars [7,8].

For RUL prediction, deep learning has been proven to have strong scalability and general-purpose capabilities to handle massive high-dimensional data, thereby realizing RUL prediction from such data [9]. Ali et al. [10] introduced the WD (Weibull distribution) to the SFAM (simplified fuzzy adaptive resonance theory mapping) neural network to predict the RUL of rolling bearings. Using an artificial neural network unit (ANN) with multiple hidden layers, Strušnik et al. [11] constructed a simulation model to predict system performance based on real data. Kim et al. [12] proposed a CNN-based (convolutional neural network, CNN) prediction model to reflect the correlation between RUL estimation and a health status detection process. Based on data trajectory expansion, a joint data-driven RUL prediction method using the AdaBoost regression model and the long-short term memory (LSTM) model was established by Zhu et al. [13]. Liu et al. [14] used stacked bidirectional LSTM to establish a data-based model to predict the RUL of supercapacitors. Kong et al. [15] proposed a framework combining deep convolutional neutral network (DCNN) and two-layer LSTM, used for lithium-ion battery state-of-health (SOH) estimation and RUL prediction. Zheng et al. [16] combined a multilayer LSTM unit with a standard feedforward layer to form a novel LSTM-based prediction model. At present, LSTM has attracted more and more attention for predicting the RUL of mechanical components due to the advantages of time series data modeling.

The above studies have verified the prospects of deep learning-based methods in RUL prediction. However, most of these methods are implemented with deterministic neural networks, which ultimately provide RUL point estimation. In practical applications, RUL prediction is affected by various types of prediction uncertainties, such as measurement uncertainty introduced by noise interference, model uncertainties related to the prediction model, and uncertainty conditions caused by the operation randomness [17]. Uncertainty quantification is the basis of many key decisions. If the uncertainty is not quantified, the estimated value of the RUL prediction point has difficulty providing sufficient guidance value for the maintenance strategy in practical applications [18,19].

Liu et al. [20] presented an incremental online learning strategy based on the relevance vector machine (RVM) algorithm, which achieved high prediction accuracy and considered the uncertainty. Tang et al. [21] used a modeling method based on the truncated normal distribution (TND) to estimate the degradation state of lithium batteries, which can simultaneously obtain the distribution of drift parameters and the RUL distribution by considering measurement uncertainty. The above methods are based on mathematical statistics and quantify the uncertainty to an extent. Nonetheless, the quantification methods consume the resources and time, and the RUL prediction methods still have certain limitations. Therefore, to obtain an accurate RUL estimation while considering uncertainty, it is necessary to incorporate uncertainty features into deep-learning-based methods. Ghahramani [22] stated that the Bayesian method is a promising measure of uncertainty, and Bayesian inference can be used as a learning tool to address uncertainty in deep learning. Additionally, Gal and Ghahramani [23] theoretically and experimentally validated that, for quantifying the uncertainty, the dropout mechanism can be applied to obtain a Bayesian approximation in the deep learning field.

Based on the above analysis and summary, it can be found that the current RUL prediction is faced with the dual problems that traditional deep learning methods have difficulty measuring the uncertainty and common uncertainty measurement methods are limited to adapt to RUL prediction methods. To overcome these difficulties, this paper proposes a bearing RUL prediction method based on LSTM and uncertainty quantification. First, in the data preprocessing stage, the bearing degradation characteristics are screened according to correlation, monotonicity, and robustness to provide the bearing degradation information for model training. Second, in the prediction stage, by introducing dropout into the LSTM network model, the RUL point and the nonparametric kernel density distribution can be obtained. The RUL point estimation and nonparametric kernel density distribution are combined to make decisions based on uncertainty. 

The paper is organized as follows. Following this introduction, the basic principle of LSTM are briefly presented in Section 2. In Section 3, the feature screening method and the proposed uncertainty-based RUL prediction model are introduced. Then, the verification and comparison of the proposed methods are carried out in Section 4. Finally, the conclusions are drawn in Section 5.

## 2. Background

### 2.1. LSTM Neural Network Model

The LSTM is a special variant of the recurrent neural network (RNN), which can alleviate the long-term dependence of the RNN by adding more complex cellular memory units. Figure 1 shows the basic unit structure of the LSTM model. Its basic unit is a memory block mainly formed by a memory cell and three gate control units (including a forget gate, an input gate, and an output gate) [24]. The memory unit is represented by the horizontal straight line at the top of Figure 1, which is used to receive information from the previous moment and transfer the processed information to the next moment. The three gate units are marked by the dashed box. They all have the same structure and consist of a sigmoid activation function and a multiplication operator in sequence. The sigmoid activation function is used to output a number in (0, 1), and the multiplication operator is used to control the throughput of other information.

### 2.2. Dropout

Uncertainty quantification is the basis of many key decisions. RUL prediction without uncertainty quantification is usually inaccurate and unreliable. Dropout, proposed by Hinton [25], is mainly used to solve the overfitting problem of machine-learning-based models. Specifically, during training the model, the network weights are randomly deleted or reduced, and the connections between the neural nodes are deleted or weakened to reduce the overfitting of the network model. 

In addition to solving the problem of overfitting, dropout can also be used to quantify uncertainty. It has been verified that the Bayesian approximation of uncertainty quantification in deep learning can be realized [26]. Bayesian theory can be introduced into deep neural networks to obtain the probability distribution of network parameters and eventually obtain an uncertainty assessment of the prediction results. However, for uncertainty quantification under the Bayesian framework, variational inference needs to be used to approximate the posterior distribution, which greatly changes the structure of the original neural network. It introduces many additional parameters and increases the computational cost. Dropout, originally used to prevent overfitting in a neural network, can be equivalent to the variational inference process in a Bayesian framework. It can also quantify the prediction uncertainty without greatly changing the neural network model used. Notably, the dropout process is similar to training several different neural networks in parallel in a given neural network. Thus, the uncertainty introduced by dropout is the model uncertainty, which is also called epistemic uncertainty.

## 3. RUL Prediction Method

The RUL prediction method based on LSTM and uncertainty quantification is proposed in this paper. As shown in Figure 2, the prediction framework mainly includes three parts: data preprocessing, model training, and RUL prediction. In the initial data preprocessing stage, the features of vibration signals are extracted and screened by a novel comprehensive evaluation index. Then, the features are normalized to reduce the scale difference between them to reduce the subsequent model training time. Next, the sample data that meet the input requirements of LSTM need to be constructed by the sliding time-window method. Finally, the sample data are divided into the training and testing sets. The training set is utilized to train the established LSTM model to determine and optimize the model parameters during the model training stage. Followed by the RUL prediction stage, the trained LSTM model with operating dropout can process the sample data to obtain the prediction results, and apply nonparametric kernel density estimation to obtain the kernel distribution of the bearing RUL by considering the uncertainty, thus providing a basis for uncertain decisions. 

### 3.1. Data Preprocessing

#### 3.1.1. Build an Alternative Feature Set

The degradation features of bearings in different domains can be extracted by different methods. Bearing degradation is a non-stationary dynamic process with strong time correlation, while frequency domain features are usually statistical features based on discrete Fourier transform, which have no advantage in reflecting the change of specific frequency over time. Thus, to capture the transient characteristics in the bearing degradation, time-frequency domain features, such as wavelet packet energy, are selected as the alternative feature sets. Additionally, the alternative feature sets contain ten kinds of common time-domain degradation features: square root mean (SRM), the root mean square (RMS), absolute mean (AM), absolute maximum (MA), skewness (Skw), kurtosis (Kur), crest factor (Cf), shape factor (Sf), clearance factor (Clf), and impulse factor (If).

In subsequent experiments, ten time-domain degradation features are calculated using the bearing acceleration signals in the horizontal and vertical. Then, four-level wavelet packet decomposition is performed to obtain 10 time-domain degradation features and 16 wavelet packet energy features. This means that for each bearing data set, the corresponding alternative feature set contains 52 alternative features.

#### 3.1.2. Evaluation and Screening of Feature Sets

In contrast to its fault diagnosis role, RUL prediction needs to fit the degradation process, which is conditioned by failure modes. Therefore, some alternative features that are only applicable to specific failure modes are not suitable for RUL prediction. Three feature evaluation indexes [27], including a correlation indicator, Corr(f,t), monotonicity indicator, Mon(f), and robustness indicator, Rob(f), are used to screen the features that can effectively reflect the degradation process and have predictability. 

Before evaluating the alternative features, the central moving average method is utilized to treat the alternative feature, f, as a random process, which is divided into a trend part, $f_{T}$, reflecting the average trend and a random part, $f_{R}$, reflecting the residual, as in Equation (1).
(1)$$f\left(t_{k}\right)=f_{T}\left(t_{k}\right)+f_{R}\left(t_{k}\right)$$
where $f\left(t_{k}\right)$ is the degradation feature value at time $t_{k}$. Based on Equation (1), the above three evaluation indexes are given by
(2)$$Corr\left(f,t\right)=\frac{\left|K\sum_{k=1}^{K}\left(f_{T}\left(t_{k}\right)t_{k}\right)-\left(\sum_{k=1}^{K}f_{T}\left(t_{k}\right)t_{k}\right)\cdot \sum_{k=1}^{K}t_{k}\right|}{\sqrt{\left[K\sum_{k=1}^{K}f_{T}^{2}\left(t_{k}\right)-{\left(\sum_{k=1}^{K}f_{T}\left(t_{k}\right)\right)}^{2}\right]}\cdot \left[K\sum_{k=1}^{K}t_{k}^{2}-{\left(\sum_{k=1}^{K}t_{k}^{}\right)}^{2}\right]}$$
(3)$$Mon\left(f\right)=\frac{1}{K-1}\left|\sum_{k=1}^{K}h\left(f_{T}\left(t_{k+1}\right)-f_{T}\left(t_{k}\right)\right)-\sum_{k=1}^{K}h\left(f_{T}\left(t_{k}\right)-f_{T}\left(t_{k+1}\right)\right)\right|$$
(4)$$Rob\left(f\right)=\frac{1}{K}\sum_{k=1}^{K}\exp\left(-\left|\frac{f_{R}\left(t_{k}\right)}{f\left(t_{k}\right)}\right|\right)$$
where K is the total number of observations and h(t) is a step function which is defined as
(5)$$h\left(t\right)=\left\{ \begin{array}{l} 1,\;t>0\\ 0,\;t\leq 0 \end{array}\right.$$

A single feature evaluation index can only evaluate partial applicability of the alternative features for RUL prediction. To comprehensively evaluate the performance of alternative degradation features, a novel comprehensive evaluation index, Ce, based on the above three single indexes is established in this paper, which is given by
(6)Ce=0.2Corr(f,t)+0.5Mon(f)+0.3Rob(f)

The weight value before each feature evaluation indicator represents the importance of the indicator. The weight value is limited to the range (0, 1). During the process of bearing degradation, damage is accumulated, thus the monotonicity of degradation features is the most important. Secondly, in order to ensure the reliability of prediction results, the robustness of degradation features should also be prioritized. Therefore, weights of 0.2, 0.5, and 0.3 were selected for Corr, Mon, and Rob, respectively. 

#### 3.1.3. Standardizing Features and Constructing Samples

The relative magnitude of different feature scales makes a big difference in training. If screened features were not standardized, then the features with large magnitudes would play a major role in model training. However, the features with small magnitudes have difficulty facilitating the update of model parameters. This means that some useful features do not participate in the model training process, which makes it difficult for the model to be trained to an optimal state. Furthermore, this would also cause repeated oscillations in the gradient direction during the model optimization process, which would slow down the convergence speed and increase the training time. Normalization of input features into a machine learning model is a standard preprocessing step and using mean and standard deviation is the most common method. Therefore, the Z-score standardization criterion is applied to process screened features. The Z-score standardization criterion is formulated as
(7)$$X_{\mathrm{stand}}=\frac{X_{orig}-\mu }{\sigma }$$
where $X_{\mathrm{stand}}$ and $X_{orig}$ are the standardized signal and the original signal, respectively, and σ and μ are the variance and mean of the original signal, respectively.

In addition to standardizing the feature signals, this paper also uses the sliding time-window method to construct sample data that meet the input requirements of LSTM. For the LSTM prediction model, the sequence length and input dimension need to be set in the model input. The sample data after processing in the previous steps are a two-dimensional array of size $N_{\mathrm{sample}\_\mathrm{point}}\times N_{\mathrm{feature}}$, where $N_{\mathrm{sample}\_\mathrm{point}}$ and $N_{\mathrm{feature}}$ are the number of sampling points and features, respectively. The sample length corresponding to the sequence length needs to be determined first. The sample length, $N_{\mathrm{length}}$, corresponding to the sequence length is set to 25 in this paper based on experience.

In order to construct samples, the length of the time window is set to $N_{\mathrm{length}}$, the data corresponding to the sampling points 0–$N_{\mathrm{length}}$ are taken as an input sample, and the RUL of the $N_{\mathrm{length}}$-th sampling point is set as the sample label. Then, the interception window is moved backward by one time unit along the sampling time axis, and the above operation is repeated to obtain a series of input samples. There is overlap between adjacent samples constructed by sliding time windows, which can make LSTM fully fit the training data.

### 3.2. Prediction Model Based on LSTM and Uncertainty Quantification

In this paper, the proposed LSTM-based prediction model consists of an LSTM layer, a fully connected layer, and a dropout layer. The LSTM layer is used to extract hidden temporal information. It also mines the temporal information in depth by superimposing multiple LSTM layers. Generally, the number of LSTM layers is set to 1–4, while this paper is set to 2. The number of LSTM layers is set according to the influence of different layers on the final result, as shown in Table 1. The fully connected layer is used to transform the hidden layer space output by the LSTM layer into the sample label space. According to the universal approximation theorem (UAT), two fully connected layers with enough hidden elements and at least one active function with a ‘squeeze’ function can fit any continuous function. Therefore, the LSTM layer is followed by two fully connected layers and the activation function is a tanh function. The dropout layer is used to prevent overfitting and quantify the prediction uncertainty in the testing stage. In this paper, a dropout layer is added between two LSTM layers and two fully connected layers. The specific prediction model parameters are given in Table 2. Furthermore, the mean squared error (MSE) cost function given by Equation (8) is adopted to evaluate the accuracy of the predicted value during model training. The Adam optimizer is adopted to minimize the cost function, and the corresponding learning rate is set to 0.0005. The parameters of the model mentioned above are roughly selected according to experience, and then optimized based on grid search method. Table 1, Table 3, Table 4 and Table 5 show the final results of the proposed model under different parameter values, in which all parameters except those with variable values are optimal.
(8)$$MSE=\frac{\sum_{i=1}^{n}{\left(f\left(x\right)-y\right)}^{2}}{n}$$
where y denotes the target value, f(x) denotes the predicted value, and n is the total number of predicted values.

Additionally, for the proposed prediction model, combining dropout and nonparametric kernel density estimation are used to quantify the uncertainty, thus obtaining the RUL point estimations and kernel distributions. As discussed in Section 2.2, the dropout process is similar to training several different networks in parallel under a given network structure. Therefore, an LSTM-based prediction model with operating dropout can obtain some different prediction results when the same data are input into the prediction model many times, which means that there is uncertainty quantification. The mean value can be regarded as the point estimation of the RUL. The nonparametric kernel density estimation can be used to process those prediction results to obtain the kernel density distributions of the RUL at different sample points, which can be used to make convincing uncertainty-based decisions.

## 4. Experiment and Results

### 4.1. Data Set Description

The PHM2012 dataset provided by the FEMTO-ST research institute in Besançon, France, contains bearing life-test data that is utilized to study the prediction performance of the bearing RUL prediction method. The experimental platform is shown in Figure 3.

In the PHM2012 dataset, the data represent a deep-groove ball bearing’s operation, and the data are mainly obtained under three different load conditions. The corresponding revolving speeds and radial loads are 1500 rpm and 5000 N, 1650 rpm and 4200 N, and 1800 rpm and 4000 N, respectively. For each load condition, there are 3–7 different bearing degradation data subsets; two data subsets are the bearing lifecycle data, and the remaining subsets are the truncated partial degradation data. Each subset mainly includes temperature signals and acceleration vibration signals in the vertical and horizontal directions. For the acceleration signal, the sampling frequency is 25.6 kHz, the duration of a single sample is 0.1 s, and the sampling interval is 10 s.

### 4.2. Experimental Setup and Model Training Process

In the PHM2012 data set, the bearing lifecycle data subsets, Bearing1_1 and Bearing1_2, are assigned to training sets, and subset Bearing1_3, which is also the bearing lifecycle data, is assigned to the testing set. The first label ‘1’ denotes the data for the first load condition (revolving speed 1800 rpm and radial load 4000 N). The second label ‘1’ (or ‘2’ or ‘3’) denotes that the duration of accelerated bearing degradation testing is 1 h (or 2 h or 3 h). The bearing acceleration signals in the vertical and horizontal directions can be used to calculate 10 time-domain degradation features. Then, 10 time-domain degradation features (RMS, SRM, AM, MA, Skw, Kur, Sf, Cf, If, and Clf) and 16 wavelet packet energy features (WPNE-01–16) are obtained by conducting four-level wavelet packet decomposition. The raw data of training set Bearing1_1 are shown in Figure 4, and 52 alternative features are obtained after processing, as shown in Figure 5. In the figure, H and V denote the features in the horizontal and vertical directions, respectively.

For each subgraph of Figure 4, the horizontal axis denotes the number of data sampling points, and the vertical axis denotes the amplitude of the corresponding degradation feature. The order of magnitude of the time-domain degradation features Sf, Cf, and If is 101 and that of the other 7 degradation features is 102–103. In addition, the order of magnitude of the wavelet packet energy is up to 105. This shows that the order of magnitude of different features is quite different, which is not conducive to parameter convergence during model training. Moreover, not all the features in the alternative feature set can effectively reflect the bearing degradation process. Hence the feature data need to be screened and standardized before being input into the prediction model.

The comprehensive evaluation indicators are calculated with Equation (6) based on the 52 alternative features of the Bearing1_1 training set. The results in descending order are shown in Figure 6. The comprehensive evaluation indicators of RMS, SRM, and AM in both the horizontal and vertical directions are large, and they show a good degradation trend close to the bearing degradation process. Eventually, based on Bearing1_1 and Bearing1_2 training sets, the total comprehensive evaluation indicators are sorted in descending order, as shown in Table 6. The time-frequency domain characteristics H-WPNE-04 and H-WPNE-03 reflect adjacent frequency bands, but H-WPNE-04 is superior to H-WPNE-0 in the comprehensive evaluation indicator of the two bearings. In order to reduce the redundancy of the selected features, H-WPNE-04 was selected and H-WPNE-03 was ignored. The features corresponding to the first nine comprehensive indicators are used as bearing degradation features to train the subsequent prediction model. The indicators are denoted as H-SRM, H-AM, H-RMS, V-SRM, V-AM, V-RMS, H-WPNE-04, H-WPNE-09, and H-WPNE-01.

After the features are evaluated and screened, their data need to be standardized by the Z-score standardization criterion. Additionally, in each bearing subset, there are 2803 sampling points. To meet the input requirements of the RUL prediction model, the length of the time window is set to 25, and the sliding time-window method is used to construct 2779 samples. All model training and testing are performed in a PyTorch deep learning framework. The training epoch is set to 200 based on experience.

### 4.3. Results Analysis

#### 4.3.1. RUL Predictive Analysis

After model training, the testing dataset needs to be utilized to validate the performance of the proposed prediction model. In the prediction stage, it is necessary to turn off the automatic reverse derivation function of the model to avoid updating the model parameters and improving the prediction speed. The dropout layer also needs to work normally to quantify the prediction uncertainty. For this paper, the testing set is input to the prediction model 1000 times, and the mean value and standard deviation of the predicted results at any time need to be obtained. Figure 7 shows the RUL prediction results, while the mean value is viewed as the point estimation of the RUL. The solid blue line represents the real RUL value, the solid red line represents the point estimation of the RUL, and the light green area denotes the distribution range of the RUL prediction results for the 1000 data inputs.

It can be seen from Figure 7 that the point estimation results can be divided into three stages: the first stage consists of the first 800 sample points, the second stage consists of the 800–1700th sample points, and the third stage consists of the last 650 sample points. For the first stage, the point estimation values of the RUL are lower than the real values and fluctuate greatly. However, the overall trend of the prediction curve tends to be horizontal, with only a slight downward trend. This is because the first stage corresponds to the early stage of the bearing running, which is still in a healthy condition, and the actual performance state does not change over time. Therefore, the RUL point estimation value does not have a clear declining trend over time. Additionally, during model training, the RUL label is set to decline linearly over time. It means that the trained prediction model cannot accurately estimate the RUL based on the first stage in which the bearing is in good condition. As a result, there is a large fluctuation in this stage. For the second stage, the RUL point estimation values are close to the real values with less fluctuation. This shows that the proposed RUL prediction method has good prediction accuracy. The stable wear stage of the bearing operation opportunely corresponds to this stage, along with the irreversible stable degradation over time of various performance indicators. Then, the precise RUL point estimation is captured by the prediction model. For the third stage, the point estimation values of RUL present irregular fluctuation with a large fluctuation range, but the overall trend still tends to the real value. Figure 7 shows that the RUL point estimation drops suddenly at the beginning of the third stage. This is because the bearing has a serious failure at this time, illustrating that the bearing has entered the accelerated degradation stage. Some features change drastically under the influence of bearing faults, causing prediction results based on these features to change significantly. Hence, the proposed RUL prediction method based on data preprocessing and LSTM can be utilized to accurately predict the bearing RUL.

In addition to showing the RUL point estimation, Figure 7 also shows the distribution range of the prediction results for 1000 data inputs. The distribution range is large in the early stage, and its upper boundary is close to the real RUL value. The middle stage distribution is narrow and evenly distributed near the real value of the RUL. The distribution range in the late-stage expands to the maximum value of the whole useful lifecycle. As can be seen in Figure 8, the degree of fluctuation and the width of the confidence interval is consistent with those of the point estimation of the RUL. Additionally, the variation trend of the 95% confidence interval width first shows a slight decrease and then a faster increase over time. This indicates that the uncertainty is the lowest in the second stage. Then, the corresponding uncertainty gradually increases.

The width variation of each sample point with a 95% confidence interval can reflect the uncertainty variation of the predicted RUL value to some extent, but it cannot be used directly to make the uncertainty-based decision. The variation trends of nonparametric kernel density distributions with samples in different stages are shown in Figure 9, Figure 10 and Figure 11, which are used to make convincing uncertainty-based decisions on the RUL prediction results in this paper.

The kernel distribution of the RUL prediction results for 800 samples in the first stage is shown in Figure 9. The point estimations deviate greatly from the real value, and the kernel distribution is scattered. This illustrates that the proposed prediction method has poor prediction performance in the first stage. However, the overall variation trend of the point estimations in the first stage is relatively regular. The RUL prediction value within a period has some reference significance, but the uncertainty is large. Figure 10 shows the kernel distribution of the RUL prediction results for 900 samples in the second stage. The point estimations agree with the real value, and the kernel distribution is concentrated, which shows that the prediction performance in the second stage is excellent. Considering that the predicted RUL values in the second stage are distributed in a straight line where the real RUL decreases linearly with time, the model obtains not only an accurate RUL prediction value at this stage, but also an accurate final useful life end time by analyzing the downward trend of the RUL prediction value over a period. Therefore, uncertainty-based decisions for the RUL prediction results can be conducted in the second stage. As seen in Figure 11, the prediction performance in the third stage is poor. The point estimations deviate greatly from the real value, and the kernel distribution is scattered. What is worse, the overall variation trend of the point estimations in this stage is irregular along with high uncertainty. The reason for the deviation between point estimation and the real value is the lack of information mining capabilities of the model itself and the insufficient information about selected features in this stage. The deviation may also be caused by the sudden failure, which cannot be considered in the prediction model. The discrete kernel distribution in this stage finally leads to increasing uncertainty due to the lack of training samples with sufficient information.

#### 4.3.2. Comparison and Discussion

To show the effect of the feature screening step in the proposed RUL prediction model, an extra RUL prediction case is conducted by replacing the feature screening with random features. Furthermore, the CNN model is used to replace the LSTM model to predict the RUL, further illustrating the superiority of the LSTM model over other networks in temporal feature extraction. The prediction results for the above different cases are shown in Figure 12.

It can be seen that the prediction result of RUL of LSTM with random features is basically maintained at about 0.5 before sample No. 1600, and drops sharply at sample No. 1600. This is because some features focus more on reflecting the accelerated degradation time of faults rather than describing the degradation process of bearings. This fully shows that feature screening is effective and necessary for RUL prediction. As the prediction model based on CNN also includes feature selection, the corresponding prediction results can reflect the bearing degradation process at an early stage. However, when the bearing enters the state of accelerated degradation (after 1600 samples), the prediction results of the model based on CNN show drastic fluctuations, which is obviously inferior to the performance of the method proposed in this paper. For the above three different situations, the corresponding mean absolute error (MAE), root mean square error (RMSE), and training time are shown in Table 7. The RMSE and MAE of the proposed model are both smaller than those of the other two models. Meanwhile, the training time is close to that of the LSTM model with random features and far lower than that of the CNN model. This further verifies the effectiveness of feature screening in improving the stability and accuracy of prediction, and the superiority of LSTM in mining hidden information in time series data compared with other traditional neural networks

To further verify the proposed method, we compared our proposed method with models proposed by adaptive Kalman filter model. Figure 13 shows the comparison of the predicted values of the model proposed in this paper and the adaptive Kalman filter model proposed by Wang et al. [29]. Table 8 shows RMSE and MAE of the two models. It can be found that the proposed method can track more complex and hidden degradation behaviors, thus the life prediction results are more accurate. In conclusion, the bearing uncertainty prediction model based on LSTM and uncertainty quantization proposed in this paper can predict the bearing uncertainty stably and accurately, providing a basis for uncertain decision-making.

## 5. Conclusions

Traditional deep-learning-based RUL prediction methods often lack the ability to quantify related uncertainties in engineering practice. Considering this, this paper proposes a bearing RUL prediction method based on LSTM and uncertainty quantification. The uncertainty quantification is introduced by combining dropout and nonparametric kernel density estimation. The effectiveness and feasibility of the proposed prediction model is validated on the PHM2012 dataset, and based on the prediction result. Some conclusions are given:

(1) The prediction results on the PMH2012 dataset illustrate that the proposed prediction model can accurately and stably predict the bearing RUL. Especially in the stable degradation stage, it has excellent prediction performance, which has the significance in practical application.

(2) The novel comprehensive evaluation index based on correlation, monotonicity, and robustness is proposed to improve the accuracy and stability of the prediction model. The process mainly focuses on evaluating and screening the features from the built alternative feature set to select the effective degradation features. Furthermore, the superiority of this strategy is perfectly validated by comparison with the LSTM-based prediction model trained with random features.

(3) The uncertainty existing in the actual prediction process is introduced by combining dropout and nonparametric kernel density estimation in this paper. The LSTM model with operating dropout can obtain RUL distribution result approximating Bayesian inference. These data are then processed using nonparametric kernel density estimation to obtain the kernel distribution that expresses uncertainty. As a result, the specific result and variation trend of point estimation and Gaussian kernel distribution of the bearing RUL can be observed on the basis of the proposed prediction model. It provides an important and scientific basis for uncertainty-based decisions.

## Figures and Tables

**Figure 1 sensors-22-04549-f001:**
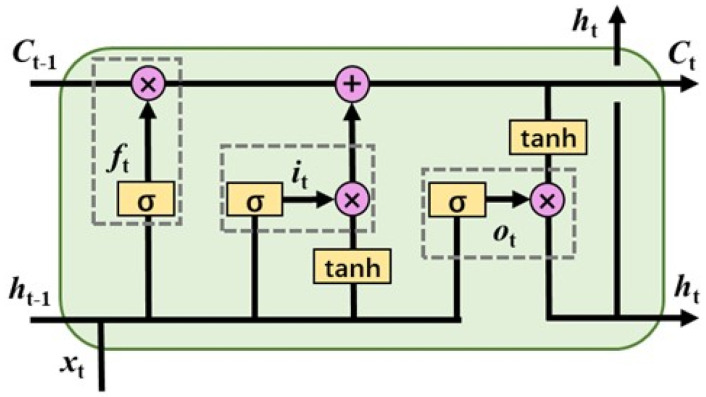
Long−short term memory (LSTM) unit structure.

**Figure 2 sensors-22-04549-f002:**
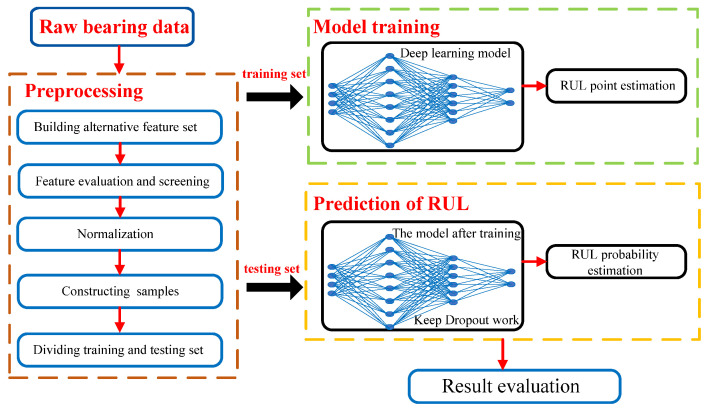
Flow chart of the bearing RUL prediction method based on LSTM and uncertainty quantification.

**Figure 3 sensors-22-04549-f003:**
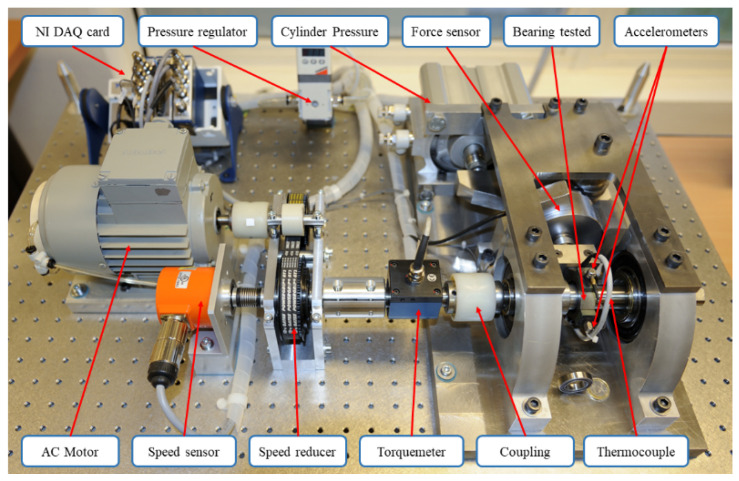
Experimental platform of bearing faults for the PHM2012 dataset [28].

**Figure 4 sensors-22-04549-f004:**
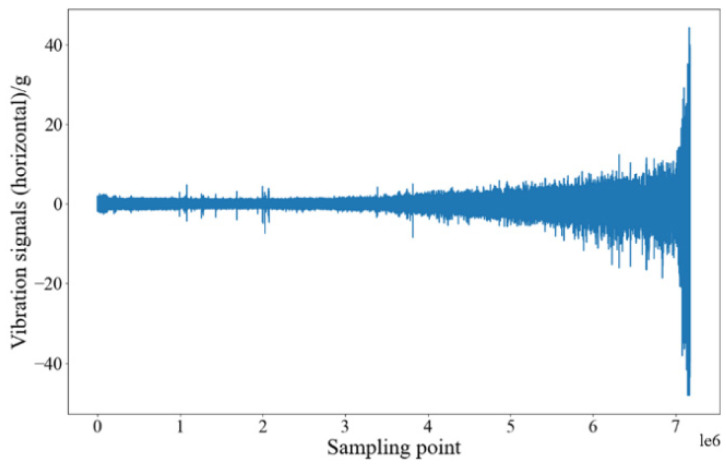
Raw Data of Bearing1_1.

**Figure 5 sensors-22-04549-f005:**
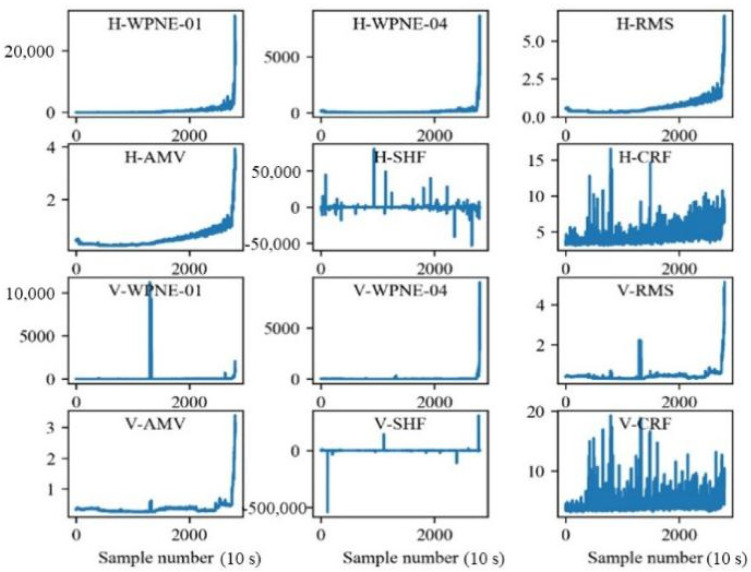
Some alternative features of Bearing1_1.

**Figure 6 sensors-22-04549-f006:**
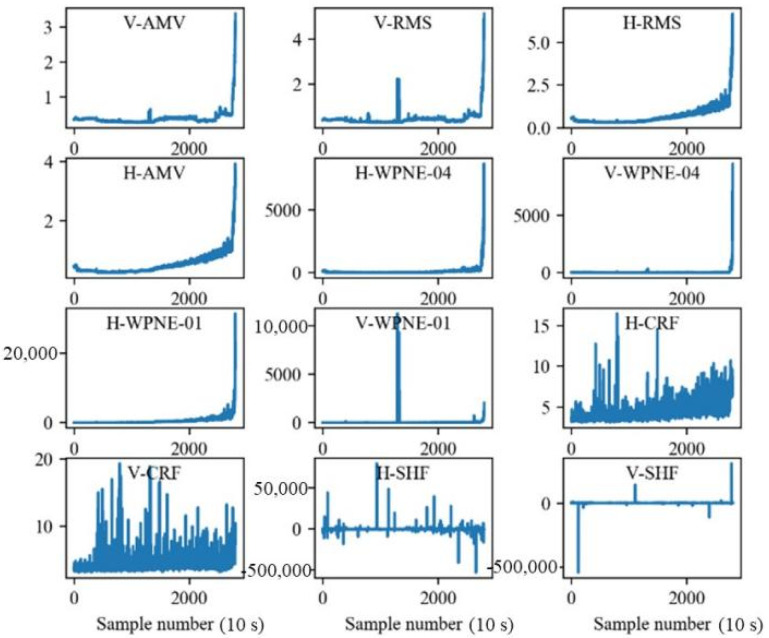
Comprehensive evaluation indicators in descending order for Bearing1_1.

**Figure 7 sensors-22-04549-f007:**
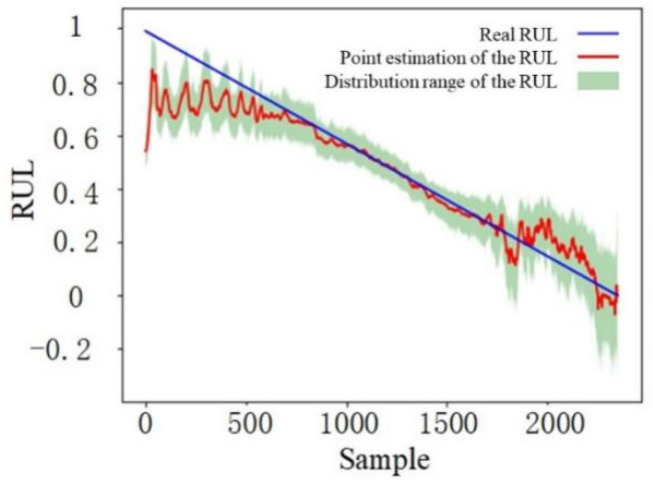
Bearing RUL prediction results.

**Figure 8 sensors-22-04549-f008:**
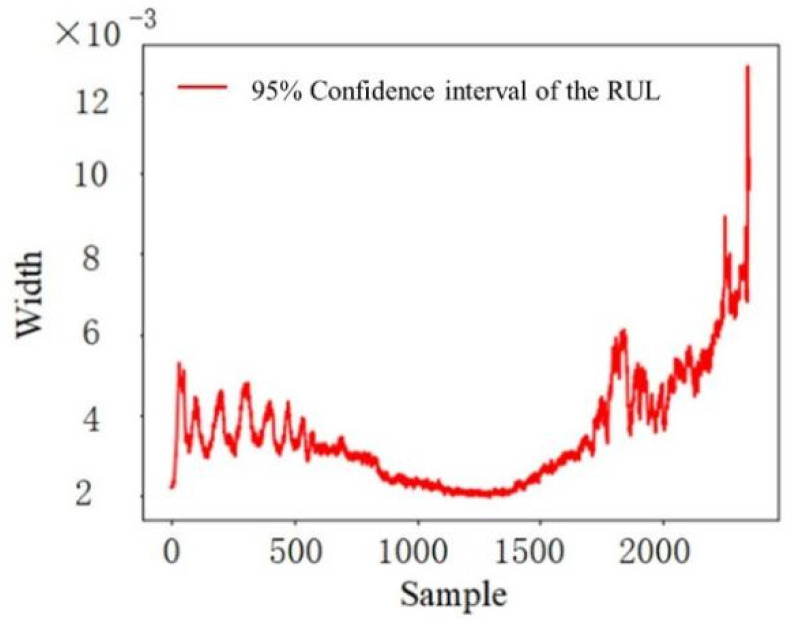
Distribution range of each sample point with a 95% confidence interval.

**Figure 9 sensors-22-04549-f009:**
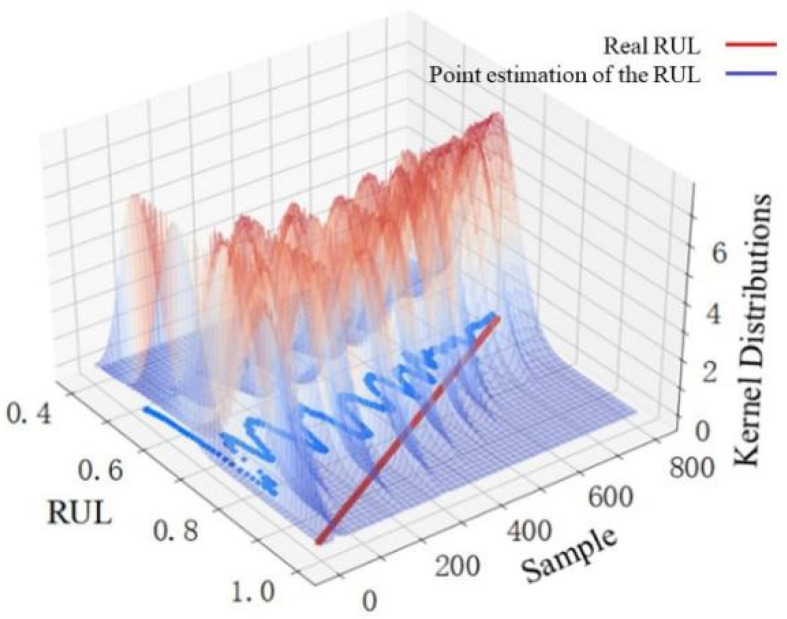
RUL point estimation and nonparametric kernel density distribution in the first stage.

**Figure 10 sensors-22-04549-f010:**
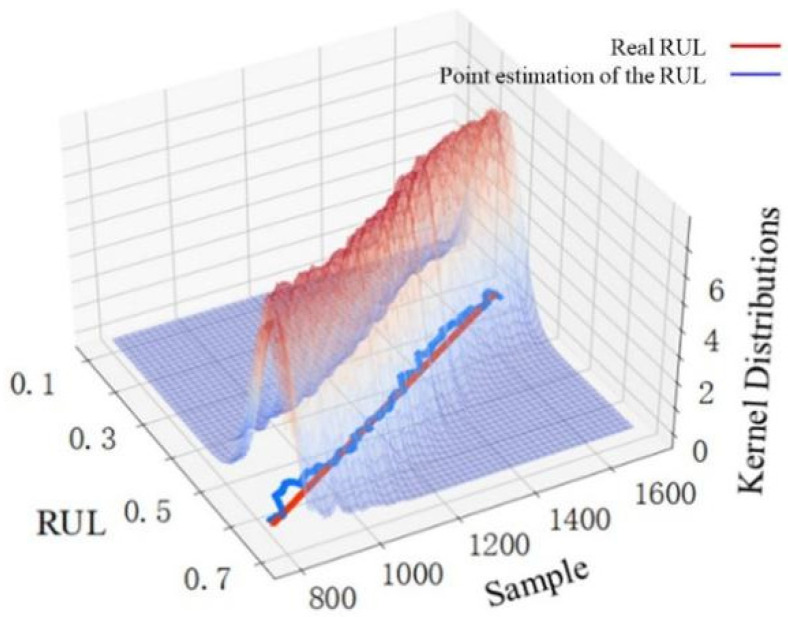
RUL point estimation and nonparametric kernel density distribution in the second stage.

**Figure 11 sensors-22-04549-f011:**
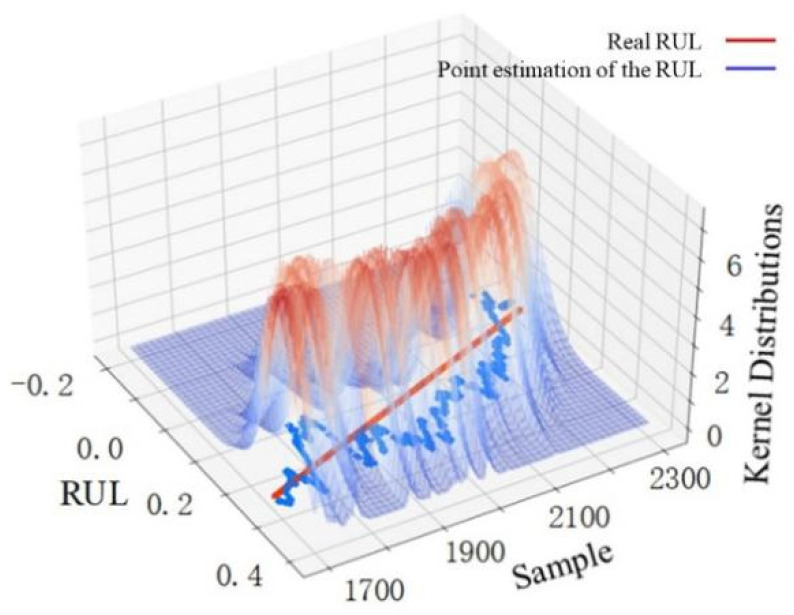
RUL point estimation and nonparametric kernel density distribution in the third stage.

**Figure 12 sensors-22-04549-f012:**
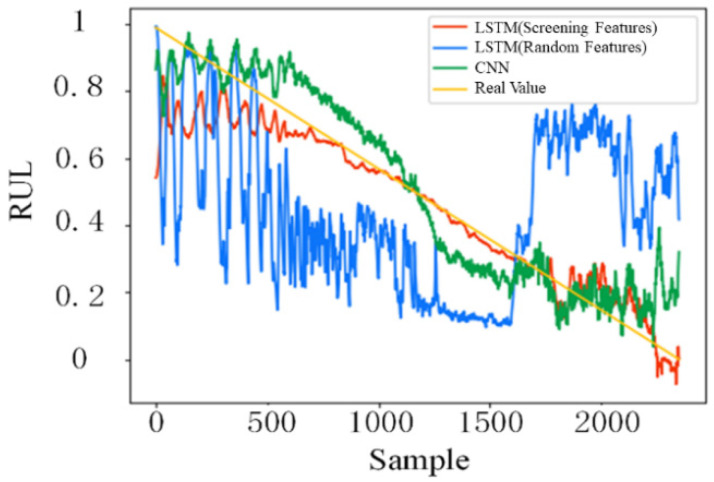
RUL prediction results for different cases.

**Figure 13 sensors-22-04549-f013:**
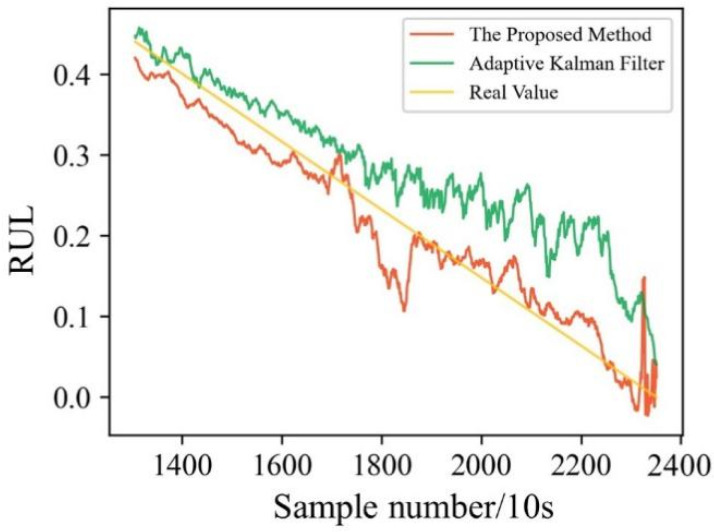
RUL prediction results for different methods.

**Table 1 sensors-22-04549-t001:** Results of the proposed method with different hide layer size and number of LSTM layers.

No.	Hidden_Size	RMSE	MAE	No.	Num_Layers	RMSE	MAE
1	50	0.0885	0.0669	1	1	0.1070	0.0890
2	100	0.0706	0.0554	2	2	0.0567	0.0385
3	200	0.0567	0.0385	3	3	0.0793	0.0582
4	300	0.0897	0.0614	4	4	0.0805	0.0536
5	400	0.1124	0.0891	5	5	0.2615	0.1575

**Table 2 sensors-22-04549-t002:** Structure of LSTM prediction model.

No.	Layers	Parameters
1	Input	(64 × 25 × 9)
2	LSTM	(9, 9)
3	Dropout	0.5
4	LSTM	(9, 9)
5	Dense	(9 × 25, 100), linear
6	Dropout	0.5
7	Dense	(100, 1), linear, tanh
8	Output	(64, 1)

**Table 3 sensors-22-04549-t003:** Results of the proposed method with different epochs and learning rates.

No.	Epoch	RMSE	MAE	No.	Learning Rate	RMSE	MAE
1	50	0.0885	0.0669	1	0.0001	0.1167	0.0805
2	100	0.0706	0.0554	2	0.0005	0.0567	0.0385
3	200	0.0567	0.0385	3	0.001	0.0699	0.0395
4	300	0.0897	0.0614	4	0.005	0.0707	0.0467
5	400	0.1124	0.0891	5	0.01	0.0976	0.0697

**Table 4 sensors-22-04549-t004:** Results of the proposed method with different batch size and length of sequences.

No.	Batch_Size	RMSE	MAE	No.	Length of Sequence	RMSE	MAE
1	16	0.1180	0.0826	1	10	0.0830	0.0550
2	32	0.0907	0.0610	2	20	0.0784	0.0537
3	64	0.0567	0.0385	3	25	0.0567	0.0385
4	128	0.0869	0.0738	4	30	0.0741	0.0566
5	256	0.0881	0.0713	5	40	0.0940	0.0736

**Table 5 sensors-22-04549-t005:** Results of the proposed method with different dropout and linear layer output size.

No.	Dropout	RMSE	MAE	No.	Linear Size	RMSE	MAE
1	0.2	0.0952	0.0675	1	10	0.1013	0.0677
2	0.35	0.0828	0.0570	2	50	0.0825	0.0564
3	0.5	0.0567	0.0385	3	100	0.0567	0.0385
4	0.65	0.0698	0.0459	4	200	0.0844	0.0542
5	0.8	0.0802	0.0667	5	400	0.0868	0.0600

**Table 6 sensors-22-04549-t006:** Total comprehensive evaluation indicators.

Name of Feature	Ce of Bearing1_1	Ce of Bearing1_2	Total Ce
V-SRM	0.292921596	0.310558419	0.603480015
H-SRM	0.289652167	0.309138323	0.59879049
V-AM	0.293166977	0.299596696	0.592763673
H-RMS	0.290015165	0.297871662	0.587886827
H-AM	0.288404719	0.298055497	0.586460216
V-RMS	0.290772079	0.269228607	0.560000686
H-WPNE-04	0.273594185	0.285045446	0.558639631
H-WPNE-03	0.267427595	0.274241517	0.541669112
H-WPNE-01	0.271408761	0.266538666	0.537947427
H-WPNE-09	0.281029679	0.256802251	0.537831929

**Table 7 sensors-22-04549-t007:** Comparison of different prediction models.

Method	RMSE	MAE	Time
The Proposed Method	0.0567	0.0385	116.35
LSTM with Random Features	0.2463	0.1864	110.17
CNN	0.1613	0.1194	590.08

**Table 8 sensors-22-04549-t008:** Comparison with other authors’ methods.

Method	RMSE	MAE
The Proposed Method	0.0567	0.0385
Adaptive Kalman Filter	0.0775	0.0640

## Data Availability

Not applicable.

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
