# Peer review of "Remaining Useful Life Prediction Method for Bearings Based on LSTM with Uncertainty Quantification"

_sensors, 2022, doi:10.3390/s22124549_

Round 1

Reviewer 1 Report

Presented in manuscript problem considers very important, practical issue. The evaluation of the life prediction of machine elements is the fundamental in the exploitation process of the engineering equipment. The new, combined method to evaluate the remaining useful life of examined rolling bearings is proposed. Paper is written consequently, logically, clearly.

I recommend to accept manuscript for publishing. In order to a little enrich a notation I propose to:

  • do the discussion of the presented in equation (6) numbers,
  • change the first, long sentence in conclusion point – lines 419-423,
  • remove the dot – line 189.

Reviewer 2 Report

Journal Sensors (ISSN 1424-8220)

Manuscript ID sensors-1704353

Type Article 

Title: Remaining Useful Life Prediction Method for Bearings Based on LSTM and Uncertainty Quantification

This paper focuses on the remaining useful life prediction method for bearings based on LSTM and uncertainty quantification. The concept is interesting, the methodology is well presented but the paper needs some major revisions. My comments:

  1. The highlights, nomenclature, parameters, subscripts, and superscripts should be added.
  2. The abstract should be revised. In the abstract, striking sentences emphasizing the work should be added. The originality of the study should be emphasized.
  3. The novelty/originality shall be further justified that the manuscript contains sufficient contributions to the new body of knowledge. The knowledge gap needs to be clearly addressed in the Introduction.
  4. In the manuscript please clarify what mean LSTM.
  5. In section 3. RUL prediction method, please add before and after pre-processing data diagram or filtered and non-filtered data obtained by the measurement.
  6. In the validation section please, compare model results and real process data obtained by the measurement. Errors should be presented.
  7. Use the same pattern for the Fig. name, bolt, etc.
  8. In Fig. 6, 7, 8, 9, 10 diagrams please mark the names of the curves.
  9. In the text, there are errors in English, that needs to be carefully read and corrected.
  10. A literature survey is not sufficient to present the most updated for further justification of the originality of the manuscript. You should carry out a thorough literature survey of papers published in a range of top energy journals so as to fully appreciate the latest findings and key challenges relating to the topic addressed in your manuscript and to allow you to more clearly present your contribution to the pool of existing knowledge. The following paper can be added: https://doi.org/10.1002/er.7375

Reviewer 3 Report

Overall, an interesting paper, with good amount of detail and some exceptionally nice graphics. Below are some questions and comments for authors’ consideration:

Line 144-145: Bearing degradation is a process with strong temporal correlation, and frequency-domain features have difficulty reflecting the changes in specific frequencies over time.

This is a bit controversial statement because some of the most successful engineered condition indicators (CIs) for bearing fault detection and tracking were obtained using enveloping and monitoring so-called bearing frequencies (cage, inner race, outer race, ball), which are the function of bearing geometry (see, e.g. Bechhoefer “A Quick Introduciton to Bearing Envelope Analysis”). Please consider softening your statement.

Line 159-160: In contrast to its fault diagnosis role, RUL prediction needs to fit the degradation process. Therefore, some alternative features that are only applicable to specific failure  modes are not suitable for RUL prediction.

The statement is a bit confusing.  Degradation mode is conditioned by the failure mode – physically, different failure modes are governed by different failure mechanism. Please clarify what was meant above.

Notation nitpicking: Heaviside step function is typically denoted by h(t), while delta symbol is reserved for the distribution, dirac’s delta function (which is a derivative of Heaviside’s step function). Please consider using standard signal processing notation.

Please consider providing some rationale for the weights in Eq. (6). Were the weights optimized in some way? The novelty seems slightly overstated if it is reduced to somewhat arbitrary weighted sum of existing evaluation indices. Although it was cited, it would be helpful to explain how are individual evaluation indices are employed to evaluate features.

Line 177-187 -- on standardization

Normalization of input features into a machine learning model is a standard preprocessing step and using mean and standard deviation is probably the most common way to do it.  It would be helpful to state that explicitly.

LSTMs have been very popular in the recent years (although somewhat overshadowed by transformers).  Researchers have shown that their simpler versions,  gated recurrent networks (GRUs) are just effective.  Did the authors consider GRUs? Why using more complex models?

Figure 3 – probably needs permission from the original authors

Figure 5: Eqs (2-4) and (6) suggest that the evaluation indices are scalars, why were they not compared as scalars? Why plotting the features instead? Please clarify the process.

Why were the training epochs set to 200 in advance? Was this arbitrary? How was the training error evolved?

Figure 6: consider adding a legend to denote traces

Interpretation of the RUL evolution. According to the traditional probabilistic interpretations of RUL (e.g. Engel), the uncertainty decreases in time, as the potential for better correspondence between feature and damage growth increases. Yet, Figure 7 shows quite the opposite.  Please consider explaining your results to these conceptually reasonable expectations from the traditional expectations of RUL probabilistic interpretations (that the uncertainty decreases as the degradation increases).   

Using dropout for model uncertainty is interesting, but it would be helpful to explain how this approach approximates Bayesian inferencing, beyond just showing that with many models, come many distributions.

Table 2: it would be helpful to compare the results obtained in this paper to the best known results obtained on the same data set from the literature (the dataset was published 10 years ago).

Reviewer 4 Report

The manuscript no. 1704353 evaluates a method based on LSTM and uncertainty quantification to predict remaining useful life of bearings. The paper has a good structure and is generally well written, however some improvements are necessary for publication. 

1. On line 171, please remove the parantheses and the bullet placed inside. 

2. A reference might be given on line 197 related to the experience mentioned there.

3. The main issue of the paper is that the parameters of the proposed method were not varied. The model was pre-configured but it was not optimized for this specific application. Thus, the variation of the number of LSTM layers would be useful (now it is set to 2 according to line 208) or, at least, the number of neurons should be varied in these layers. The variation of the learning rate around the default value 0.01 would be also necessary. The number of training epochs might be also varied. Similarly, the batch size and the dropout rate must be varied. The Conclusions section should present the optimal configuration of the proposed prediction method.

4. Please write "testing dataset" instead of "testing set data" on line 299.

5. A legend might be included into figure 6.

Round 2

Reviewer 2 Report

Thank you for inviting me to review the manuscript below:
Title: Remaining Useful Life Prediction Method for Bearings Based on LSTM and
Uncertainty Quantification

The paper life prediction method for bearings based on LSTM and uncertainty quantification.
The concept is interesting and the methodology is well presented. The results are clearly given
and adequately discussed. The paper may be publishable as an original paper after minor
revisions.

1. Where possible, in the diagrams please indicate the coordinate units.

2. Graphic abstract can be added.

3. Please compare results with other authors in this field.

4. The English language can be checked

Reviewer 4 Report

The parameter optimization problem is still unsolved. The results of the grid search based parameter optimization might be included as tables or charts showing from the perspective of each varied parameter that the proposed configuration is indeed the best one.
